

# Fecal glucocorticoid metabolite and T3 profiles of orphaned elephants differ from non-orphaned elephants in Zambia

Daniella E. Chusyd[1], Janine L. Brown[2], Steve Paris[2], Nicole Boisseau[2], Webster Mwaanga[3], Moses Kasongo[3], Lisa Olivier[3], Stephanie L. Dickinson[4], Bailey Ortyl[4], Tessa Steiniche[1], Steven N. Austad[5,6], David B. Allison[4,5] and Michael D. Wasserman[7]

[1] Department of Environmental and Occupational Health, Indiana University, Bloomington, IN, United States of America
[2] Center for Species Survival, Smithsonian's National Zoo and Conservation Biology Institute, Front Royal, VA, United States of America
[3] Game Rangers International, Lusaka, Zambia
[4] Department of Epidemiology and Biostatistics, Indiana University, Bloomington, IN, United States of America
[5] Nathan Shock Center, University of Alabama—Birmingham, Birmingham, AL, United States of America
[6] Department of Biology, University of Alabama—Birmingham, Birmingham, AL, United States of America
[7] Department of Anthropology and Human Biology Program, Indiana University, Bloomington, IN, United States of America

Corresponding author
Daniella E. Chusyd, dchusyd@iu.edu

## ABSTRACT

**Background**. Elephants provide valuable insight into how early-life adverse events (ELAEs) associate with animal health and welfare because they can live to advanced ages, display extensive cognitive and memory capabilities, and rely heavily on social bonds. Although it is known that African savanna elephants that experienced ELAEs, such as being orphaned due to human activities, have altered behavioral outcomes, little is known regarding the physiological consequences associated with those stressors.

**Methods**. We compared fecal glucocorticoid (fGCM) and thyroid (fT3) metabolites as well as body condition scores (BCS) in rescued and rehabilitated orphaned (early-dry season: $n = 20$; late-dry season: $n = 21$ elephants) African savanna elephants in Kafue National Park, Zambia to age- and sex-matched wild non-orphaned controls groups (early-dry season: $n = 57$; late-dry season: $n = 22$ elephants) during the early- (May/June) and late- (September/October) dry seasons, respectively. Age and sex were known for orphans. For non-orphan controls, age was estimated based on dung diameter, and sex was determined based on external genitalia. Hormone concentrations were compared between groups by age class to account for developmental and nutritional transitions experienced in early life. Given that environmental stressors (*e.g.*, availability of food and water sources) change over the course of the dry season, early- and late-dry seasons were separated in the analyses.

**Results**. fGCM concentrations were higher in orphans at younger ages than non-orphaned controls of any age. This may be due to the younger orphans being temporally closer to the traumatic event and thus not having had sufficient time to establish meaningful social bonds that could buffer the negative outcomes associated with ELAEs. Alternatively, orphans could have acclimated to living under human care, resulting in fGCM concentrations that were not different from wild controls at older ages. Orphans

also had significantly higher mean fT3 concentrations than non-orphans, suggesting increased caloric intake during rehabilitation. There was no difference in BCS between orphan and non-orphan elephants at any age or time period, possibly reflecting the limitations associated with BCS assessments in younger elephants.

**Conclusions.** Together, these results provide insight into possible physiological responses underlying ELAEs and/or living under human care, including alterations in fGCM and fT3 concentrations, particularly in younger orphans. While these hormonal changes suggest a physiological response to trauma, the support of social bonds and acclimation to human care may mitigate long-term stress effects, highlighting the critical role of social integration in elephant rehabilitation and conservation efforts.

## INTRODUCTION

People and animals experiencing early life adverse events (ELAEs) are at increased risk of developing health problems later in life and having reduced lifespans (*Afifi et al., 2008*; *Felitti et al., 1998*; *Tung et al., 2016*; *Snyder-Mackler et al., 2020*; *Danese & McEwen, 2012*; *Smith et al., 2008*; *Schalinski et al., 2016*; *Chu et al., 2013*). With growing evidence that negative health outcomes follow ELAEs, a better understanding of long-term physiological consequences might lead to improved interventions to minimize the negative effects. ELAEs are hypothesized to be linked to compromised later-life health due to allostatic load (*McEwen, 2007*). Allostatic load refers to the cumulative 'wear and tear' on the body due to stressors across the life span, especially stressors experienced during sensitive developmental windows, such as human childhood (*McEwen & Seeman, 1999*). As an adaptive system, the hypothalamus coordinates a cascade of complex interactions between multiple systems in response to stressors. Behavioral and neuroendocrine responses (hormonal, autonomic) begin with the release of primary stress mediators, such as corticotropin-releasing hormone, adrenocorticotropin hormone, and cortisol *via* the hypothalamic-pituitary-adrenal (HPA) axis (*McEwen & Seeman, 1999*). The HPA axis interacts with other endocrine axes, including the hypothalamic-pituitary-thyroid (HPT) axis. Therefore, the HPT axis is also a stress-sensitive system (*Anifantaki et al., 2021*), with thyroid hormone concentrations altered by allostatic load leading to thyroid dysregulation (*Chrousos & Gold, 1992*). Adding to the complexity of the stress response is that experiencing ELAEs while the brain is developing can exert a programming effect on particular neuronal networks in the brain that then influence the HPA axis (*McEwen, 2007*). Thus, experiencing a traumatic event during early life can leave a phenotypic imprint—one that may be adaptive to help the individual survive to adulthood and reproduce, but that may also be costly and often detrimental to later life morbidity and mortality.

In social, nonhuman mammals, ELAEs have primarily been investigated in the context of caregiver (*e.g.*, loss of a parent, rearing experience), social (*e.g.*, social status), or ecological (*e.g.*, drought) adversity (*Dettmer & Chusyd, 2023*). Historically, ELAE

studies were conducted primarily on laboratory-housed nonhuman primates (*Sanchez, 2006*; *McCormack et al., 2022*; *Dettmer, Suomi & Hinde (2014)*; *Dettmer et al., 2017*), but recently, there has been an expansion to include domestic animals (*e.g.*, dogs, pigs) and wild populations (*e.g.*, baboons, gorillas, chimpanzees, African savanna elephants) (*Pierantoni, Albertini & Pirrone, 2011*; *Gimsa et al., 2022*; *Zipple et al., 2021*; *Tung et al., 2016*; *Goldenberg & Wittemyer, 2018*; *Morrison et al., 2023*; *Girard-Buttoz et al., 2021*; *Lee et al., 2022*). Particular importance has been placed on understanding the loss of an offspring's mother given the mother's role in protection, resource allocation, social status, and social and emotional development (*Dettmer et al., 2017*; *Pryce et al., 2004*; *Lu et al., 2019*). The increased interest in ELAEs highlights the recognized importance of understanding its role on lifelong health and fitness outcomes.

African savanna elephants (*Loxodonta africana)* live in matriarchal societies, with families comprised of genetically related adult and subadult females and their immature offspring. Savanna elephants are nutritionally dependent on their mother for the first 2–4 years of life, display slow developmental stages, are socially dependent for approximately 10–16 years, and are capable of living into their 70s (*Lee & Moss, 1999*). During the birthing process, mothers are tended to by other females, and the entire group shares in the care of calves (*Moss, Croze & Lee, 2011*). Young calves are constantly touched and guided throughout the first years of life. Thus, the family is critical for the development, security, and learning process of young elephants. Disruption in the attachment process, including maternal separation, may have lifelong physiological and social consequences.

There was a resurgence in African elephant poaching for ivory during the early 2000s, peaking around 2011–2012 (*Underwood, Burn & Milliken, 2013*; *Chase et al., 2016*; *CITES, 2021*), with some populations declining by 30% (*Chase et al., 2016*). Poachers generally target older elephants because of their larger tusk size (*Wittemyer, Daballen & Douglas-Hamilton, 2013*; *Chiyo, Obanda & Korir, 2015*), many of which are reproductively-active females that leave behind orphaned calves and juveniles. In addition, in some areas, habitat fragmentation and loss have accelerated human-elephant conflict (HEC). Elephants enter communities to feed on crops, leading to perceived or real social and economic costs, reduced food security, and a risk to the farmers' physical wellbeing (*Kiffner et al., 2021*). This conflict often results in retaliatory killings and aggressive behaviors towards elephants (*Mariki, Svarstad & Benjaminsen, 2015*), which can also lead to calves being orphaned. Although orphaned weaned calves may be incorporated into related or nonrelated herds (*Wittemyer, Daballen & Douglas-Hamilton, 2013*), others that are not yet weaned, or those that become separated and lost from the herd, cannot survive without human intervention.

Orphaned elephants between the ages of 3 to 8 years in Kenya have an 86% probability of survival compared to 97% for non-orphans (*Parker et al., 2021*). Of those that do survive, evidence supports a range of behavioral ramifications, from neutral to detrimental, of becoming an orphaned elephant. For example, orphaned elephants appear to have different social partners compared to non-orphaned elephants, with decreased access to adult elephants (*Goldenberg & Wittemyer, 2017*). Although these orphans have a similar number of social partners, the authors hypothesized that the lack of access to mature adults may have resource acquisition and fitness consequences. On the more extreme end is the

development of violent behavior. Specifically, in South Africa, male elephants orphaned because of poaching were translocated to other areas; years later, the teenage bulls were uncharacteristically violent in killing rhinoceroses (*Bradshaw et al., 2005*). Male elephants were described as displaying post-traumatic stress disorder (PTSD)-like hyperaggressive behaviors believed to contribute to 90% of all male elephant deaths in the community, compared with 6% in relatively unstressed areas (*Bradshaw et al., 2005*). Thus, it is clear that there were lasting behavioral effects of ELAEs in this population, but less is known about the physiological consequences.

In Zambia, like many other African countries, poaching and HEC have left young elephants orphaned, often requiring subsequent rescue and rehabilitation for a chance at survival, such as that coordinated between Zambia's Department of National Parks and Wildlife and Game Rangers International. These elephants offer a unique opportunity to examine the physiological ramifications of experiencing such an early life trauma. We compared fecal glucocorticoid (fGCM) and thyroid hormone (fT3) metabolite concentrations and body condition scores (BCS) in 21 orphaned elephants to age- and sex-matched non-orphaned elephants as controls during the early- and late-dry seasons of 2021. We expected orphaned elephants to have elevated fGCM from greater physiological and psychological stress, elevated fT3 due to extra food-provisioning and stress dysfunction, and greater BCS due to extra food-provisioning and seemingly less walking compared to control elephants.

## MATERIALS & METHODS

### Ethics statement

This study was approved by the Indiana University Animal Use and Care Committee (22-022) and Zambia's Department of National Parks and Wildlife (DNPW; NPW/8/27/1).

### Animals and setting

This study was in collaboration with Game Rangers International (GRI) and Zambia's DNPW. GRI is a nonprofit that works with DNPW to rescue orphaned elephants to ultimately rehabilitate and release them back into the wild in Kafue, Zambia. Life stages of all elephants and reasons for orphaning are shown in Table S1. All orphans were estimated to be less than 24 months of age at the time of rescue and were initially housed at a nursery in Lilyai Reserve, Lusaka, Zambia. At the age of approximately 36–48 months, each orphan was translocated to a release facility located within Kafue National Park (herein referred to as 'Kafue'), Zambia. Elephants engage in a soft release, deciding when they return to the wild (*e.g.,* increased interactions with wild elephants and changes in behavior indicating a preference for remaining out in the bush rather than returning to their enclosure, with the ultimate decision of not returning to the enclosure with the rest of the orphans). The timing of this decision varies, although it typically does not occur before 168 months (7 years) of age. Dietary supplementation for orphans is outlined in Table S2. Regardless of location (*i.e.,* Lilayi Nursery or Kafue Release Facility), orphans were in the bush browsing from approximately 7:00–12:00 hr and 14:00–17:00 hr, and in Kafue, were able to interact with wild elephants. Control elephants residing in Kafue were located within approximately

11 miles of the release facility and located based on known elephant hotspots (*i.e.,* an area where elephants either spend a significant amount of time or regularly visit) and were included for comparison. The local elephant population typically remains within 15 miles of the facility year-round primarily due to permanent water sources and ample resources. The age of control elephants was estimated based on visual assessment of body size, physical development, the eruption of tusks, length and circumference of tusks, body shape and proportions, and dung diameter (Fig. S1) (*Moss, 1996*; *Morrison et al., 2005*). The primary criteria selection for the control elephants was based on their estimated age, such that they were of the same approximate age as the orphaned elephants (∼16–186 months).

## Data collection

In 2021, data were collected during the early-dry (May–June) and late-dry (September–October) seasons. The early-dry season was defined as the first two months (May/June) following the last rains, while the late-dry season was defined as the last two months (September/October) before the start of the rains. The inclusion of seasonal time points was intended to account for the changes in food and water sources from the beginning to the end of the dry season. At the time of dung collection, orphaned elephants either resided at the nursery or release facility or were already released back into the wild in Kafue (Table S1). Sex was assigned based on genital and morphological differences (*Moss, 1996*).

In the early-dry season, samples were collected from all 20 orphaned elephants (Table S1). During the late-dry season, samples were collected from 20 of 21 orphaned elephants as one released orphaned elephant was not located for sample collection, and one new orphan was rescued during the late-dry season and included only in the latter time point. Demographics of the orphans are outlined in Table S1. We collected 57 and 22 samples from control elephants in the early- and late-dry season, respectively, of which 14 and 15 came from identified control individuals (*i.e.,* the elephant was observed defecating and associated data collected), respectively. Demographics of the control elephants by season are outlined in Table S1.

## Dung collection

For orphaned and wild elephants, fecal samples were collected within 4 h of defecation, as previously described (Fig. 1) (*Webber et al., 2018*). Time of defecation was based on observed defecation, observation of the arrival of the focal herd (*i.e.,* the research team observed the elephant herd arrive, and thus, the dung collected was likely not there prior to the herd's arrival), and dung characteristics (*i.e.,* shiny, minimal to no bug disturbance, no other animal disturbance). In brief, the diameter of two intact dung balls was measured with a flexible measuring tape and averaged per side (Fig. S1). The average was used to estimate age (*Morrison et al., 2005*) even for identified individuals. Then, ∼300 g (2 large handfuls) of feces were collected from the middle of each dung ball, placed in a sterile, labeled bag, and put in a cooler until arriving at the field site. Dung was mixed by hand for a minimum of 2 min and then stored frozen in the field freezer (−20 °C). Each dung sample received a certainty score, indicating the degree of certainty that that dung sample came from a specific individual (scores 0–3; a score of 3 represents observed defecation; Table S1), and the corresponding dung sample was then collected.
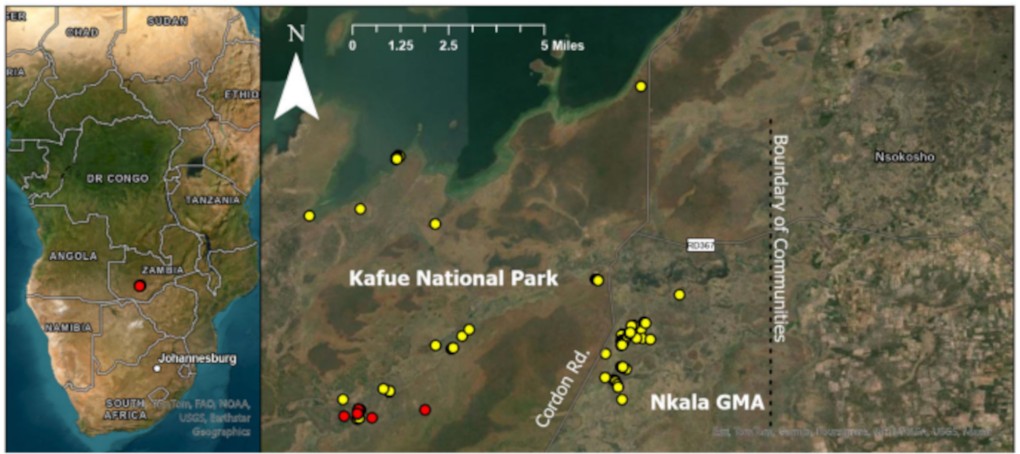

**Figure 1** **Location of each dung sample collected from orphans residing at KRF and wild controls.** Red dots represent orphaned elephants and yellow dots represent wild controls. Cordon road divides Kafue National Park from Nkala game management area (GMA), which is a buffer zone created between the national park and communities to the east (located within the black box). The location of communities is displayed to provide context regarding the proximity between humans and our wild control elephants as these elephants are known to engage in crop raiding. All wild control dung samples were collected within 11 miles of the orphans.

Within 12 months, samples were removed from the freezer and thawed, and then placed on a solid-phase extraction (SPE) cartridge (Hypersep C8 SPE column 500 MG/3 ml, Product # 60108-309; Thermo Fisher Scientific, Waltham, MA, USA), adapted from *Edwards et al. (2014)*. In brief, approximately one g ($\pm 0.01$ g) of each sample ($N = 119$) was weighed (careful consideration was taken to avoid weighing any large undigested plant material) and placed into a 15-ml falcon tube. Four ml of 90% ethanol was added to each tube and rotated for 60 min at 30 rpm. Samples were then centrifuged for 30 min at 4,000 rpm. The supernatant was poured into a clean tube and centrifuged for 10 min at 1,400 g. Then, 6.75 ml of ultra-pure water was added to achieve a final loading concentration of 40%. Cartridges were primed by pushing through four ml of deionized water at one ml/minute using a 60 ml syringe, followed by four ml of 100% ethanol at one ml/minute. The fecal sample was then loaded onto the cartridge at a rate of one ml/2 min. Then two ml of deionized water was pushed through at a rate of one ml/min using a 60 ml syringe. The cartridge was sealed with parafilm and stored in a dark place at ambient temperature until shipment to Indiana University (IU) for hormone analyses.

### Body condition score

Video was recorded of each elephant to assign a body condition score (BCS) (score 1–9) adapted from *Fernando et al. (2009)* and *Morfeld et al. (2014)*, based on the prevalence of the backbone, hips, and ribs. BCS was assigned to 15 and 14 orphans during the early- and late-dry seasons, respectively. For control elephants, when possible, BCS was assigned to the elephant that defecated (early-dry season: $n = 12$; late-dry season: $n = 9$). However, when it was unclear which elephant the dung came from, BCS was assigned to an elephant

of the same age class from the same herd associated with the collected dung (early-dry season: $n = 25$; late-dry season: $n = 2$).

## Hormone analysis

At IU, SPE cartridges were eluted using a vacuum manifold, slowly passing through five ml of 100% ethanol. The extract was dried down using an Evap-o-Rac, and then reconstituted in one ml 80% ethanol. Extracts were pipetted into microcentrifuge tubes and stored at −80 °C until assay analyses. Concentrations of fGCM were analyzed by a double-antibody enzyme immunoassay (EIA) validated for elephants (*Watson et al., 2013*). Standards (0.078–20 ng/ml; Sigma Diagnostics, St. Louis, MO, USA), samples (diluted 1:5–1:31 in assay buffer; X065 Arbor Assays, Arbor, MI, USA), and quality control samples (50 µl/well) were added in duplicate to pre-coated goat anti-rabbit IgG, 96-well plates at room temperature. Corticosterone-HRP (25 µl; 1:25,000; C. Munro, University of California, Davis, CA) was added to all wells, followed by 25 µl rabbit anti-corticosterone antibody (CJM006; 1:60,000; C. Munro) to all but non-specific binding wells. Plates were sealed and incubated at room temperature over a period of 1 or 2 h for corticosterone and fT3, respectively. Unbound components were removed by washing five times with wash buffer (X007, Arbor Assays, Ann Arbor, MI, USA), followed immediately by the addition of a chromagen solution containing TMB (100 µl, X019, Arbor Assays, Ann Arbor, MI, USA) to each well. After incubation for 30 min at room temperature, the reaction was halted by adding stop solution (50 µl; X020 Arbor Assays, Ann Arbor, MI, USA) and absorbance was determined at 450 nm with a reference of 630 nm. Concentrations of fT3 in fecal extracts (diluted 1:16, 1:24, or 1:26 in assay buffer) were determined using a validated commercial double-antibody EIA (Catalog #K056-H5; Arbor Assays, Ann Arbor, MI, USA) (*Szott et al., 2020*). Assay sensitivities were 0.02 ng/ml and 0.04 ng/ml for corticosterone and fT3, respectively. fGCM and fT3 intra- and inter-assay coefficients were 4.2% and 6.6%, and 4.6% and 8.1%, respectively. All hormone results are expressed as ng/g dry weight (DW).

## Data analysis

SAS (SAS 9.4) was used for all analyses. Descriptive statistics were conducted by group and by season. Normality was assessed by skewness, and fGCM and fT3 were log-transformed for analysis. To assess whether there were differences in fGCM or fT3 concentration, or BCS between orphans and controls, separate linear mixed models (LMM) using restricted maximum likelihood (REML) were conducted using hormone concentration or BCS as the outcome variable and random effects for each individual to account for correlation of repeated measures across the two seasons within-animal. Fixed effects included group (orphans or control), age (as a continuous variable), and season (early-dry or late-dry). Predicted marginal means were then broken down by approximate age class, corresponding to the youngest age, transition from infant to juvenile (∼48 months), juvenile to subadult (∼96 months), and then midway through sub adulthood (∼137 months) to create another subcategory as our study population was relatively young, with all individuals except one being a subadult or younger. These classifications were used to represent the elephant's developmental and transitional milestones. As hormone concentrations have

been documented to differ by age and season in other populations (*Wasser et al., 2010*), interactions for group x age and group x season were also included. In other species, the relationship of age on fGCM and fT3 is not always linear (*Lupien et al., 2005*; *Gopinath et al., 2016*), thus quadratic terms were also considered and then included where significant ($p < .05$). Sex was unknown for a subset of the control elephants. All models were run separately on only the animals with known sex to test for differences in sex. While the models were run on log-transformed data for the two hormones to adjust for non-normally distributed residuals, means and confidence intervals were back-transformed to the original scale for interpretations. Significance was determined where $p < 0.05$ (two-tailed).

## RESULTS

Overall fGCM concentrations by group and by season are presented in Table 1. In linear mixed models with a random effect for individual, there was a significant interaction for group by age (estimate: 0.003, CI [0.001–0.005], $p = 0.018$) such that younger orphans had higher fGCM concentrations compared to control elephants (Table 2). For elephants aged 4 and 46 months (selected as the youngest age and the approximate age of transitioning from an infant to juvenile, respectively), orphans had estimated fGCM concentrations greater than that of similar-aged control elephants (estimate: −0.319, CI [−0.524 to −0.115], $p = 0.003$; estimate: −0.204, CI [−0.340 to −0.067], $p = 0.004$, respectively; Table 3). There were no discernible differences between groups in the older age classes ($p > 0.05$; Table 3). There was also a significant effect of season, where fGCM concentrations in the early-dry season were significantly higher than in the late-dry season (estimate: 0.118, CI [0.010–0.225], $p = 0.034$; Table 4). Concentrations of fGCM are depicted by group by age (Fig. 2A) and by group (Fig. 3A) by season. Within orphans, fGCM concentrations declined with age (Fig. 2A; estimate: −0.002, CI [−0.004 to −0.001], $p = 0.010$).

Overall fT3 concentrations by group and by season are presented in Table 1. In linear mixed models with a random effect for individual, there was a significant interaction for group by age (Table 2; estimate: −0.012, CI [−0.022 to −0.001], $p = 0.039$) such that juveniles (∼48–92 months) and younger subadult orphans (∼137 months) had higher fT3 concentrations compared to similar-aged control elephants (Table 2). There was no significant difference in fT3 between groups at other age classes ($p > 0.458$). Concentrations of fT3 by group by age are shown in Fig. 2B, with concentrations by group by season depicted in Fig. 3B. Within orphans, fT3 concentrations demonstrated a quadratic relationship with age.

Overall BCS by group and by season are depicted in Table 1. BCS was not significantly different between the groups, as the estimated mean for the BCS for orphans was 5.9, while for controls the average BCS was 6.0 (Fig. 2C, Table 3).

When analyzing only elephants of known sex, sex was not significant in any models for fGCM ($p = 0.408$), fT3 ($p = 0.068$), or BCS ($p = 0.122$).

**Table 1** Descriptive statistics for samples from orphaned and control elephants overall and by season, on original measurement scale, not log transformed.

| | N | Mean | SD | Min | Max | Median |
|---|---|---|---|---|---|---|
| **Overall hormone concentrations for orphaned and control elephants** | | | | | | |
| O.E.: fGCM (ng/g DW) | 40 | 101.5 | 59.6 | 24.9 | 298.0 | 83.0 |
| C.E.: fGCM (ng/g DW) | 79 | 93.9 | 50.0 | 14.5 | 197.3 | 88.3 |
| O.E.: fT3 (ng/g DW) | 39 | 271.8 | 123.2 | 42.7 | 502.9 | 285.1 |
| C.E.: fT3 (ng/g DW) | 79 | 98.7 | 71.9 | 12.4 | 320.2 | 78.6 |
| O.E.: BCS (1–9) | 29 | 5.9 | 1.3 | 3.0 | 8.0 | 6.0 |
| C.E.: BCS (1–9) | 48 | 6.2 | 1.3 | 3.0 | 8.0 | 6.0 |
| **Orphaned elephants during the early-dry season** | | | | | | |
| Age (months) | 20 | 93.1 | 50.0 | 16.0 | 183.0 | 93.0 |
| fT3 (ng/g DW) | 20 | 269.2 | 126.2 | 77.0 | 502.9 | 284.1 |
| fGCM (ng/g DW) | 20 | 107.3 | 67.7 | 36.7 | 298.0 | 81.7 |
| BCS (1–9) | 15 | 5.9 | 1.2 | 4.0 | 8.0 | 6.0 |
| **Orphaned elephants during the late-dry season** | | | | | | |
| Age (months) | 20 | 88.9 | 50.1 | 19.0 | 186.0 | 81.5 |
| fT3 (ng/g DW) | 20 | 274.3 | 123.3 | 42.7 | 486.9 | 285.1 |
| fGCM (ng/g DW) | 19 | 95.3 | 50.9 | 24.9 | 220.5 | 88.7 |
| BCS (1–9) | 14 | 5.9 | 1.5 | 3.0 | 8.0 | 5.5 |
| **Control elephants during the early-dry season** | | | | | | |
| Age (months) | 57 | 88.8 | 43.0 | 4.0 | 180.0 | 84.0 |
| fT3 (ng/g DW) | 57 | 88.3 | 58.2 | 12.4 | 320.2 | 74.8 |
| fGCM (ng/g DW) | 57 | 103.0 | 48.9 | 14.5 | 190.0 | 108.1 |
| BCS (1–9) | 37 | 6.4 | 1.3 | 3.0 | 8.0 | 7.0 |
| **Control elephants during the late-dry season** | | | | | | |
| Age (months) | 22 | 101.2 | 46.0 | 18.0 | 168.0 | 102.0 |
| fT3 (ng/g DW) | 22 | 125.5 | 95.4 | 18.1 | 295.6 | 86.2 |
| fGCM (ng/g DW) | 22 | 70.2 | 45.6 | 25.1 | 197.3 | 55.8 |
| BCS (1–9) | 11 | 5.6 | 1.1 | 4.0 | 7.0 | 6.0 |

**Notes.**

There were 21 orphans total. One orphaned elephant (age 158 months during the early-dry season), which was sampled during the early-dry season, was not sampled during the late-dry season because he was already released in the wild and at the time of sample collection, his GPS collar was not working. Between sample collection time points, a new elephant was rescued (age 24 months during the late-dry season) and was subsequently included in the late-dry season. These occurrences resulted in $n = 20$ at both time points and the lower mean age during the late-dry season compared to the early-dry season.

OE, orphaned elephants; CE, control elephants.

## DISCUSSION

People who experience ELAEs, such as neglect and maltreatment, parental separation, or death of a parent, are at increased risk of mental, behavioral, cardiovascular, immune function, and several other negative health outcomes across the lifespan (*Dunn et al., 2017*; *Crawford et al., 2022*; *Slopen et al., 2013*, *Suglia et al., 2020*). Similar evidence supports long-lasting physiological alterations attributed to ELAEs in other nonhuman species as well (*Dettmer & Chusyd, 2023*). Elephants are long-lived, display complex emotions and cognition, rely heavily on social bonds, and are at increased risk for experiencing ELAEs due to historical and ongoing human-elephant conflict. Thus, understanding ELAEs' role

**Table 2   Results from linear mixed models for each outcome of interest, type 3 tests.** *P*-values < 0.05 are bolded.

|  | fCGM (Log) | | | fT3 (Log) | | | BCS | | |
| --- | --- | --- | --- | --- | --- | --- | --- | --- | --- |
|  | **F** | **df** | ***p*-value** | **F** | **df** | ***p*-value** | **F** | **df** | ***p*-value** |
| Group | 9.6 | 97 | **0.003** | 0.1 | 97 | 0.789 | 0.1 | 64 | 0.819 |
| Season | 5.5 | 14 | **0.034** | 0.7 | 13 | 0.432 | 1 | 7 | 0.166 |
| Age | 2 | 14 | 0.180 | 2.9 | 13 | 0.113 | 2.4 | 7 | 0.350 |
| Group*Season | 1.5 | 14 | 0.238 | 0.4 | 13 | 0.529 | 1.3 | 7 | 0.291 |
| Group*Age | 7.2 | 14 | **0.018** | 5.3 | 13 | **0.039** | – | – | – |
| Age$^2$ | – | – | – | 1.6 | 13 | 0.224 | – | – | – |
| Group*Age$^2$ | – | – | – | 5 | 13 | **0.043** | – | – | – |

Notes.
*Note that the *p*-value for group in this table is not the overall effect but that at age 0 years.

**Table 3   Estimated marginal means by group by age, for each outcome from LMM, (back-transformed to original scale).**

|  |  |  | Orphans | | Controls | | | |
| --- | --- | --- | --- | --- | --- | --- | --- | --- |
|  |  |  | **Mean** | **(L, U)** | **Mean** | **(L, U)** | ***t*** | ***p*** |
| fGCM (ng/g DW) | age = 4 | Min | 132.4 | (92.2, 190.3) | 63.5 | (47, 85.8) | 3.1 | **0.003** |
|  | age = 46.2 | Mean - 1SD | 108.0 | (84.9, 137.5) | 67.6 | (55.3, 82.6) | 3.0 | **0.004** |
|  | age = 91.66 | Mean | 86.9 | (72.6, 104) | 72.3 | (62.8, 83.3) | 1.6 | 0.114 |
|  | age = 137.4 | Mean + 1SD | 69.6 | (54.3, 89.1) | 77.5 | (64.1, 93.7) | −0.7 | 0.492 |
|  | age = 186 | Max | 55.0 | (37.2, 81.3) | 83.4 | (61.5, 113) | −1.7 | 0.099 |
|  | ED Season |  | 92.7 | (70.7, 121.5) | 88.9 | (75.6, 104.6) | 0.3 | 0.783 |
|  | LD Season |  | 81.5 | (61.7, 107.6) | 58.8 | (45.4, 76.3) | 1.8 | 0.088 |
| fT3 (ng/g DW) | age = 4 | Min | 99.7 | (48.8, 131.6) | 78.9 | (47.3, 131.6) | −0.5 | 0.598 |
|  | age = 46.2 | Mean - 1SD | 211.1 | (157.1, 283.6) | 72.5 | (56.7, 92.6) | −5.5 | **<0.001** |
|  | age = 91.66 | Mean | 247.1 | (198.4, 307.6) | 79.2 | (66.6, 94.2) | −8.1 | **<0.001** |
|  | age = 137.4 | Mean + 1SD | 289.5 | (213.8, 391.9) | 86.6 | (213.8, 391.9) | −6.3 | **<0.001** |
|  | age = 186 | Max | 165.2 | (75.5, 361.6) | 79.2 | (66.6, 94.2) | −0.7 | 0.486 |
|  | ED Season |  | 244.2 | (174.6, 341.7) | 71.5 | (58.5, 87.4) | 6.8 | **<0.001** |
|  | LD Season |  | 249.9 | (178.6, 349.9) | 87.7 | (63.7, 120.9) | 4.9 | **<0.001** |
| BCS (ng/g DW) | age = 91.66 | Mean | 5.9 | (5.4, 6.4) | 6.0 | (5.5, 6.4) | −0.2 | 0.819 |
|  | ED Season |  | 5.9 | (5.2, 6.7) | 6.4 | (5.9, 6.9) | −1.1 | 0.295 |
|  | LD Season |  | 5.8 | (5, 6.6) | 5.5 | (4.6, 6.4) | 0.6 | 0.585 |

Notes.
*P*-values <0.05 are bolded. While the models were run on log-transformed data for the two hormones to adjust for non-normally distributed residuals, means and confidence intervals are back-transformed to the original scale for interpretations.
ED, Early-dry season; LD, Late-dry season; SD, Standard deviation; L, Lower confidence interval; U, Upper confidence interval; age is in months.

in an elephant's physiological trajectory is critical as it has implications for their fitness and survival. Our study assessed the physiological consequences of experiencing early life trauma and determined that it is associated with fGCM and fT3 concentrations by age. Specifically, infant and juvenile orphans had significantly higher fGCM concentrations compared to similar-age matched control elephants, as well as higher fGCM compared to older orphaned elephants. In addition, mean concentrations of fT3 were significantly higher in orphaned elephants compared to similar-age matched controls, and demonstrated a

**Table 4  Estimated marginal means for each outcome by season from LMMs (back-transformed to original scale).**

| | Early Dry Season | | | Late Dry Season | | | | |
|---|---|---|---|---|---|---|---|---|
| | Mean | Lower | Upper | Mean | Lower | Upper | t | p |
| fGCM (ng/g DW) | 90.8 | 77.5 | 106.3 | 69.2 | 57.3 | 83.8 | 2.3 | **0.034** |
| fT3 (ng/g DW) | 132.1 | 108.7 | 160.7 | 148.1 | 117.4 | 186.8 | −0.8 | 0.432 |
| BCS (1–9) | 6.2 | 5.7 | 6.6 | 5.7 | 5.0 | 6.3 | 1.6 | 0.166 |

**Notes.**
*P-values < 0.05 are bolded.

quadratic relationship (*i.e.,* as age increases, fT3 concentrations increase quickly at first, then slow down and plateau, before ultimately decreasing). There were no differences in BCS by orphan status. Collectively, these results suggest that ELAEs may impact an elephant's hormonal response, but the support of an adopted herd or acclimation to living under human care may attenuate the long-term physiological stress associated with experiencing trauma. The observed altered fT3 excretion by age may be associated with the trauma or an artifact of living under human care.

fGCM was higher in younger orphans compared to similar-aged control elephants and older aged orphans, suggesting elevated fCGM concentrations occur in the short-term, but over time, the orphaned elephants acclimate to their new reality, which may contribute to the observed decline in fGCM concentrations with age in the orphaned elephants. Thus, the results support the notion that, in this population, there were no long-lasting differences in fGCM between orphaned and non-orphaned elephants. Similar results, demonstrating the importance of time since becoming an orphan and potentially the role of social bonds, have been observed in female orphaned elephants that remained in the wild following maternal loss (*Parker et al., 2022*). This contrasts with our study population, where all orphans were rescued and the majority were still living under human care at the time of sample collection. Specifically, *Parker et al. (2022)* did not observe significant differences in fGCM concentrations between orphan and non-orphans living in the wild, when assessed several years after orphaning. However, in contrast to our findings, *Parker et al. (2022)* demonstrated that orphans without their natal family had lower fGCM concentrations compared to both non-orphans and orphans remaining with their natal family. The authors speculated this might be attributed to hypocortisolism due to a lack of familial support. Collectively, the studies suggest that fGCM homeostasis may be maintained through strong social bonds. One of the most robust and well-documented relationships in epidemiology is that between social connectedness and morbidity and mortality (*Holt-Lunstad, Smith & Layton, 2010*). Receiving and providing social support has been demonstrated to improve well-being by reducing the chronic activation of the stress response (*Eisenberger, 2013*); thus, social networks can play a critical role in attenuating negative outcomes following exposure to ELAEs. This may explain why younger, but not older, orphans display higher fGCM compared to similar aged control and orphaned elephants; the younger orphans are temporally closer to their traumatic event and have yet to have sufficient time to establish meaningful social bonds. It has been further demonstrated that orphans that remained with

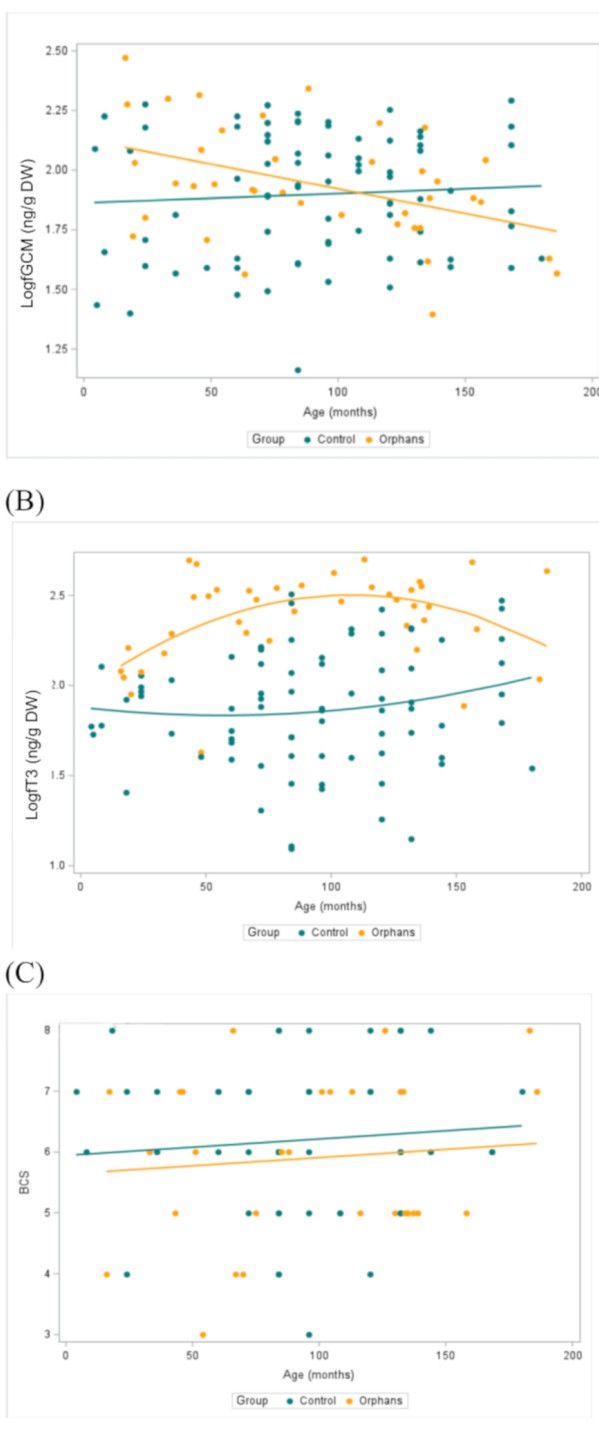

**Figure 2 Concentrations of (A) fGCM (Log10) (Control $n = 79$, Orphan $n = 40$), (B) fT3 (Log10) metabolites (Control $n = 79$, Orphan $n = 40$), and (C) BCS (Control $n = 48$, Orphan $n = 29$) by group and by age with a best fit line for orphaned and non-orphaned elephants in Zambia.** Within each group, data points are combined across seasons. ED, Early-dry season; LD, Late-dry season.

(A)

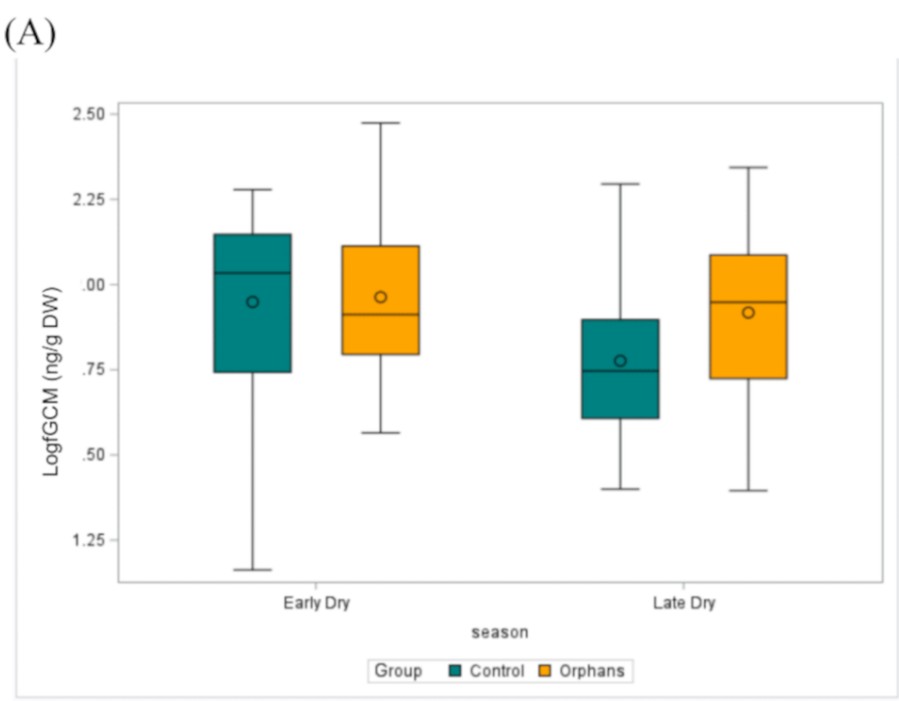

(B)

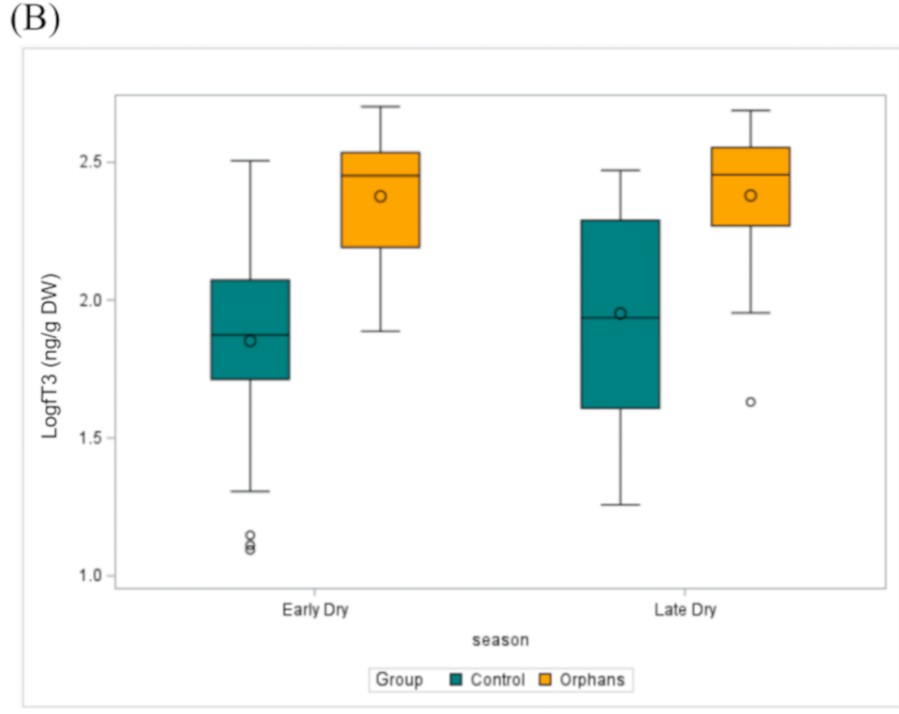

**Figure 3** **The distribution of (A) fGCM (Log10) (ED: Control *n* = 57, Orphan *n* = 20; LD: Control *n* = 22, Orphan *n* = 19) and (B) fT3 (Log10) (ED: Control *n* = 57, Orphan *n* = 20; LD: Control *n* = 22, Orphan *n* = 20) in each group for each season in the observed data.** The bottom and top edges of each box represent the range of values between the 1st and 3rd quartiles. (continued on next page...)

**Figure 3 (…continued)**
The marker inside the box represents the mean, while the line inside the box represents the median value. The whiskers indicate the range of values outside of the quartile range. Data points whose distances from the interquartile range are greater than 1.5 times the size of the interquartile range are outliers indicated by the open circles. While this plot is descriptive, linear mixed models to test for differences by group and season are described in the methods and results text. ED, Early-dry season; LD, Late-dry season.

elephant herds suffer a social cost, receiving more aggressive behaviors from conspecifics than non-orphans, and being more isolated from adult females (*Goldenberg & Wittemyer, 2018*; *Parker et al., 2021*; *Goldenberg & Wittemyer, 2017*). In this highly social animal, the role of social bonds cannot be understated, and the unique situation of our study population, where all elephants were orphaned, suggests integration and social support may be particularly important. Alternatively, it is possible that the orphans became more acclimated to their overall new environment, and living under human-care, which could also explain the decrease in fGCM with age in the orphans, which at older ages, trended to be lower than that of the control elephants, albeit not statistically significant. Ultimately, glucocorticoids are involved in numerous processes, including development, glucose levels, blood pressure, and suppression of inflammation, with a recent meta-analysis showing a strong positive correlation between glucocorticoids and metabolic rate (*Jimeno & Verhulst, 2023*). While it is unclear why fGCM concentrations decline with age in orphaned elephants, a relationship not observed in wild controls, it is likely due to consequences of being an orphan (*e.g.*, formula-fed, being alone in the wild for a duration of time without access to breastmilk, potential difference in development, different social systems, living under human-care).

Further, season had an overall effect on fGCM concentrations, such that fGCM concentrations were significantly higher in the early-dry compared to the late-dry season. Although we observed a larger difference between seasons in control compared to orphaned elephants, the difference between groups was not statistically significant. Yet, the seasonal effects may reflect different drivers for each group, such as changes in social dynamics and ecological changes. For example, for the control elephants, the majority of fecal samples were collected in a buffer zone between the national park and surrounding communities. Anthropogenic stressors (*e.g.*, being outside of protected areas, exposure to human-elephant conflict) are known to increase fGCM concentrations in elephants (*Pokharel & Brown, 2023*). In the communities around Kafue, maize is harvested between February and April, and the palm fruit is ripe by June, when crop raiding is at its peak. This may provide the elephants with more opportunities for high-energy food, thus stimulating the HPA axis and contributing to elevated fGCM concentrations. Engaging in crop-raiding activities may disproportionately contribute to the observed seasonal differences in fGCM even though none of the orphaned elephants in this study have experienced such situations. Whether the released orphans will engage in these behaviors in the future and how their physiology will respond to these environmental challenges is currently unknown, but warrants future longitudinal studies.

While fT3 concentrations were greater in orphaned elephants compared to control elephants, this relationship was driven by age class, with higher concentrations observed

for elephants between 46 and 137 months of age (juvenile to early sub-adulthood). These differences may reflect altered nutritional resources due to orphans living under human care or be attributed to altered T3 secretion due to experiencing an extreme stressor during a sensitive developmental window. In a variety of species, an increase in thyroid hormone concentration is associated with an increase in calorie intake and basal metabolic rate, with the opposite being true (*Wasser et al., 2010*; *Dias et al., 2017*; *Cristóbal-Azkarate et al., 2016*). Although there are limited thyroid hormone studies in elephants (*Szott et al., 2020*; *Wasser et al., 2010*), *Szott et al. (2020)* recently demonstrated that fT3 concentrations are positively associated with food resources. Of the 21 orphans, 19 lived under human care at the time of sample collection and received dietary supplementation year-round. This may account for elevated fT3 concentrations, particularly following sufficient time to recover from any dehydration or malnourishment associated with orphaning. This may also explain the quadratic relationship observed between fT3 and age in orphaned elephants. The rise and then subsequent plateauing of fT3 concentrations around 80-100 months may reflect orphans receiving "power balls" as additional supplementation at younger ages.

Alternatively, fT3 secretions may reflect changes associated with ELAEs. The HPT axis plays a role in regulating physiological energy demands and development (*Medici et al., 2015*) and responds to severe (but not minor) traumatic stressors (*Mason, 1968*). Numerous studies have observed elevated T3 concentrations in men who exhibit combat-related PTSD (*Mason et al., 1994*; *Kozarić-Kovačić, Karlović & Kocijan-Hercigonja, 2002*). This relationship has also been documented in women who have experienced sexual abuse during childhood (*Friedman et al., 2005*), although other studies have demonstrated an opposite relationship (*Machado et al., 2015*). Thus, it is possible that experiencing early life trauma may have altered T3 secretion patterns in the orphaned elephants, explaining why they had elevated concentrations compared to the control elephants. In people, T3 appears to play an important role in pubertal development and may start to decline after puberty, although results have demonstrated a large degree of plasticity in T3 secretion in childhood and adolescence (*Taylor et al., 2023*). If the observed secretion pattern in orphans was typical of elephant development, we would expect to see a similar pattern in our control elephants, which was not the case. The uncertainty of the fT3 profiles in orphans may be untangled as more orphans are released into the wild. If orphans living in the wild have comparable fT3 profiles to their wild counterparts, it can be deduced that living under human care artificially increases fT3 concentrations. In contrast, if their fT3 profiles remain similar to those orphans under human care, this could indicate an alteration in T3 secretion attributed to experiencing early life trauma. To our knowledge, we are the first to measure fT3 in orphaned elephants, and there is an overall paucity of published data on T3 patterns in African elephants. Our results emphasize the importance of understanding factors that impact T3 concentrations and changes across the life course.

The finding that there were no discernible differences in body condition between orphaned and control elephants, even in light of the hormonal differences, was surprising. In Asian elephants, orphaned elephants exhibit retarded growth patterns (*Weihs et al., 2001*), which, anecdotally, appears to be the case for some of the elephants in the current study population. The results may reflect our small sample size and associated lack

of power or limitations associated with body condition scoring in younger elephants. Specifically, although BCS is observed to decline with age in elephants, this is typically true for adult elephants (*Schiffmann et al., 2020*; *Chusyd et al., 2018*). The average age of the study population was only 92 months. *Schiffmann et al. (2020)* demonstrated that BCSs in infants and juveniles (<96 months of age) remain relatively stable at higher scores (6–8 out of 10) even while body mass linearly increases. Thus, it is possible that muscle and/or fat mass (subcutaneous and/or visceral) is changing within the orphans and control elephants to varying degrees, but current BCS techniques are not sensitive enough to pick up subtle differences. Rather, differences in BCS may only become apparent with extreme changes (*Chusyd et al., 2019*).

A comparison of orphaned and non-orphaned elephants presents a unique opportunity to investigate and infer the causality of ELAEs on health and welfare. In addition, studying individuals that were both temporally still quite close to the traumatic event *versus* further removed from the event allowed us to ascertain short- and long-term changes. The current study did have limitations. While some patterns can be inferred, most orphans were still under human care and only two time points were included. Longitudinal studies, following the same individuals across the lifespan, and as they transition from the orphanage into the wild, will likely provide further insights into the long-term consequences of experiencing ELAEs. Further, the sample size was comparatively small. Thus, any findings where we could not reject the null hypothesis should not be interpreted as proof that the null hypothesis is true, as a type II error is possible. This limitation may also explain why certain outcomes, including interactions, were significant when orphaned and control elephants were pooled together, but not when assessed separately. Also, sex was unknown for several control elephants. Although we ran models separately on only the animals with known sex and there was no relationship between sex and either hormone, it is possible that hormonal profiles can vary by sex. Lastly, it was unclear whether repeat samples were collected from the same control elephants. To address these, we are currently analyzing a more robust panel of biomarkers in the same orphans (plus new rescues) and using GPS-collared known control elephants, longitudinally. Including behavioral differences was outside the scope of the present study, but future work will target how physiological state relates to social networks and overall behaviors. By doing so, we will be able to account for the role of genetics and epigenetics, while creating a more holistic picture of the elephant's health and behavioral profile.

Physiological differences observed between orphaned and non-orphaned elephants regarding the association of age suggest that there are alterations associated with being an orphan, either due to alterations in hormonal secretion patterns or to living under human care. Our results also underscore the importance of social relationships for the rehabilitation of elephants. Continued research following orphaned elephants as they mature through different life stages, including leaving the orphanage, will provide essential insight into mechanisms associated with variation in adult phenotypes. Such information will be useful as orphans are likely important to population recovery following human-elephant conflict while also having the potential to improve our general understanding of how ELAEs impact an individual's health.

## CONCLUSIONS

People and animals who experienced ELAEs are known to develop later-life health issues. Thus, we hypothesized that orphaned elephants would as well, characterized by elevated fGCM from greater physiological and psychological stress, elevated fT3 due to extra food-provisioning and stress dysfunction, and greater BCS due to extra food-provisioning and seemingly less walking compared to non-orphaned control elephants. Orphans displayed different hormonal profiles, but not BCSs, compared to non-orphaned controls. Whether the differences in hormone concentrations by group were attributed to orphan status or living under human care is unclear. Future work will focus on longitudinal data following the orphaned elephants as they transition from living under human care to residing in the wild. We suspect that physiological differences will persist even once orphaned elephants return to the wild, similar to observations from other species.

## ACKNOWLEDGEMENTS

We graciously thank Zambia's Department of National Parks and Wildlife, in addition to Game Rangers International staff, keepers, and research assistants Constance Banda, Mary Muyoyeta, Kasi Kalande and former research assistant Vincent Abere, for their assistance with this project. We also thank Eric Johnson for his assistance with GIS for mapping collected samples and Kurt White for feedback on statistical analyses.

### Funding

This work was supported by the National Institute on Aging (K01AG072615), the Animal Models for the Social Dimensions of Health and Aging Research Network (NIH R24AG065172), the Morris Animal Foundation, and the Indiana Clinical and Translational Sciences Institute (EPAR1195). The funders had no role in study design, data collection and analysis, decision to publish, or preparation of the manuscript.

### Grant Disclosures

The following grant information was disclosed by the authors:
National Institute on Aging: K01AG072615.
Animal Models for the Social Dimensions of Health and Aging Research Network: NIH R24AG065172.
Morris Animal Foundation.
Indiana Clinical and Translational Sciences Institute: EPAR1195.

### Competing Interests

Daniella Chusyd is a scientific advisor to Game Rangers International. Webster Mwaanga, Moses Kasongo, and Lisa Olivier are or were employed by Game Rangers International at the time of the study and writing the original draft of the manuscript. There are no other competing interests.

## Author Contributions

- Daniella E. Chusyd conceived and designed the experiments, performed the experiments, prepared figures and/or tables, authored or reviewed drafts of the article, and approved the final draft.
- Janine L. Brown conceived and designed the experiments, authored or reviewed drafts of the article, and approved the final draft.
- Steve Paris analyzed the data, authored or reviewed drafts of the article, and approved the final draft.
- Nicole Boisseau analyzed the data, authored or reviewed drafts of the article, and approved the final draft.
- Webster Mwaanga performed the experiments, authored or reviewed drafts of the article, and approved the final draft.
- Moses Kasongo performed the experiments, authored or reviewed drafts of the article, and approved the final draft.
- Lisa Olivier conceived and designed the experiments, authored or reviewed drafts of the article, and approved the final draft.
- Stephanie L. Dickinson analyzed the data, prepared figures and/or tables, authored or reviewed drafts of the article, and approved the final draft.
- Bailey Ortyl analyzed the data, prepared figures and/or tables, authored or reviewed drafts of the article, and approved the final draft.
- Tessa Steiniche analyzed the data, authored or reviewed drafts of the article, and approved the final draft.
- Steven N. Austad conceived and designed the experiments, authored or reviewed drafts of the article, and approved the final draft.
- David B. Allison conceived and designed the experiments, authored or reviewed drafts of the article, and approved the final draft.
- Michael D. Wasserman conceived and designed the experiments, authored or reviewed drafts of the article, and approved the final draft.

## Animal Ethics

The following information was supplied relating to ethical approvals (i.e., approving body and any reference numbers):

The Indiana University Animal Use and Care and Use Committee (22-022) and Zambia's Department of National Parks and Wildlife (DNPW;NPW/8/27/1) approved this research.

## Field Study Permissions

The following information was supplied relating to field study approvals (i.e., approving body and any reference numbers):

Field research was approved by Zambia's Department of National Parks and Wildlife (DNPW;NPW/8/27/1).

## Data Availability

The SAS code and the raw data are available in the Supplemental Files.

**Supplemental Information**

Supplemental information for this article can be found online at http://dx.doi.org/10.7717/peerj.19122#supplemental-information.

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
