# Peer review of "Fecal glucocorticoid metabolite and T3 profiles of orphaned elephants differ from non-orphaned elephants in Zambia"

_PeerJ, doi:10.7717/peerj.19122_

## Round 0.1 · original submission · Minor Revisions

Dear authors, I kindly ask you to carefully make the necessary corrections to this manuscript in accordance with all the comments of the reviewers.

·

Basic reporting

The manuscript is well-written and provides a clear narrative on the physiological impacts of early-life adverse events (ELAEs) on orphaned African elephants, representing a novel study. The authors establish a strong background on ELAEs and their implications for animal health. The language style is professional, with a logical flow from introduction to conclusion. There are some limitations, notably the small sample size and the inclusion of only two time points, which the authors appropriately acknowledge in the discussion. Figures, tables, and supplementary data are generally well-presented. However, if feasible, Figure 1 and Figure 2 could benefit from additional markers or annotations to highlight significant age-related differences and enhance clarity in group comparisons.

Experimental design

The study is designed to address an important question regarding the physiological consequences of trauma and rehabilitation in orphaned elephants, and the authors employ appropriate methodologies, including fecal hormone analysis, a validated non-invasive technique.

The criteria for control selection require further clarification. While the study outlines several criteria for selecting control elephants (Lines 168–172), it remains unclear which inclusion criteria were prioritized. This information would enhance the reproducibility and interpretation of the study.

Furthermore, the fact that many control elephants are of unknown sex could influence the interpretation of hormonal differences between groups, as hormonal profiles often vary by sex. This limitation should be discussed in greater detail to strengthen the validity of the study design.

Validity of the findings

The findings provide novel insights into the physiological impacts of ELAEs, particularly in relation to fecal glucocorticoid metabolites (fGCM) and fecal thyroid hormone (fT3) concentrations.

For fGCM
For fGCM, the finding that social bonds may mitigate stress by lowering fGCM levels is compelling. A more detailed exploration of how released orphaned elephants interact with wild herds could provide further context for these findings, though this may be beyond the scope of the current study. For example, do released elephants integrate into wild herds over time, or do they remain solitary? Such social dynamics may play a critical role in stress regulation.

Additionally, the higher fGCM levels observed in the early-dry season compared to the late-dry season raise questions. While the authors attribute this to anthropogenic factors during the early-dry season, it is puzzling that late-dry conditions, with worsening food scarcity, do not intensify physiological stress responses. Could the relative abundance of crops during the early-dry season provide elephants with more opportunities for high-energy food, thereby stimulating the HPA axis and contributing to elevated fGCM levels? This hypothesis may be served as a plausible explanation.


For fT3
For fT3, the elevated concentrations observed in orphaned juvenile elephants compared to controls are noteworthy. The manuscript attributes this to dietary supplementation and possible metabolic adaptation to new conditions. This raises the question of whether human-provided food has a stronger influence on fT3 levels than ELAEs. The two orphaned elephants that were already released back into the wild during sample collection offer an opportunity to address this question. Did their fT3 profiles resemble those of other orphans or the controls? This comparison would provide insights into whether dietary supplementation has a lasting effect on fT3 levels.

Additionally, it would be valuable to discuss discrepancies between the elephant data and human studies, such as those by Machado et al. (2015; https://doi.org/10.1016/j.ijdevneu.2015.10.005), which found that ELAEs impair the conversion of T4 to T3, resulting in lower T3 levels in humans at adolescent. These differences may highlight unique aspects of thyroid hormone physiology in African elephants that are yet to be fully investigated.


For BCS
For BCS, the lack of significant changes in scores as the dry season progresses may be attributed to the test's limited sensitivity, as the authors suggest. However, it would be helpful to discuss whether alternative tools with greater sensitivity could be utilized for such studies in wildlife populations. For example, are there imaging or body composition techniques that are practical for field use and more sensitive than BCS?

Reviewer 2 ·

Basic reporting

Please see comments in the PDF.
In general, there are areas that need to be reworked for clarity and consistency.

Experimental design

Please see comments in the PDF.

Validity of the findings

Please see comments in the PDF.

Additional comments

Please see comments in the PDF.

Annotated reviews are not available for download in order to protect the identity of reviewers who chose to remain anonymous.

---

## Round 0.2 · accepted · Accept

Dear authors, I congratulate you on the acceptance of this manuscript for publication.

·

Basic reporting

No further comment

Experimental design

No comment

Validity of the findings

No comment